# Interfacial Behavior of Solid- and Liquid-like Polyelectrolyte Complexes as a Function of Charge Stoichiometry

**DOI:** 10.3390/polym13213848

**Published:** 2021-11-07

**Authors:** Hongwei Li, Martin Fauquignon, Marie Haddou, Christophe Schatz, Jean-Paul Chapel

**Affiliations:** 1Centre de Recherche Paul Pascal (CRPP), UMR CNRS 5031, University of Bordeaux, 33600 Pessac, France; hongwei.li@u-bordeaux.fr (H.L.); Martin.Fauquignon@enscbp.fr (M.F.); mariehaddou@gmail.com (M.H.); 2Laboratoire de Chimie des Polymères Organiques (LCPO), University of Bordeaux, CNRS, Bordeaux INP, UMR 5629, 33600 Pessac, France

**Keywords:** polyelectrolyte complexes, complex coacervates, liquid–liquid and liquid–solid transition, surface tension, pendant drop, interfacial tension

## Abstract

We systematically investigate in this work the surface activity of polyelectrolyte complex (PECs) suspensions as a function of the molar charge ratio Z (= [-]/[+]) from two model systems: the weakly and strongly interacting poly (diallyldimethylammonium chloride)/poly (acrylic acid sodium salt) (PDADMAC/PANa) and poly (diallyldimethylammonium chloride)/poly (sodium 4- styrenesulfonate) (PDADMAC/PSSNa) pairs, respectively. For both systems, the PEC surface tension decreases as the system approaches charge stoichiometry (Z = 1) whenever the complexation occurs in the presence of excess PDADMAC (Z < 1) or excess polyanion (Z > 1) consistent with an increased level of charge neutralization of PEs forming increasingly hydrophobic and neutral surface-active species. The behavior at stoichiometry (Z = 1) is also particularly informative about the physical nature of the complexes. The PDADMAC/PANa system undergoes a liquid–liquid phase transition through the formation of coacervate microdroplets in equilibrium with macroions remaining in solution. In the PDADMAC/PSSNa system, the surface tension of the supernatant was close to that of pure water, suggesting that the PSSNa-based complexes have completely sedimented, consistent with a complete liquid–solid phase separation of an out-of-equilibrium system. Besides, the high sensitivity of surface tension measurements, which can detect the presence of trace amounts of aggregates and other precursors in the supernatant, allows for very accurate determination of the exact charge stoichiometry of the complexes. Finally, the very low water/water interfacial tension that develops between the dilute phase and the denser coacervate phase in the PDADAMAC/PANa system was measured using the generalized Young–Laplace method to complete the full characterization of both systems. The overall study showed that simple surface tension measurements can be a very sensitive tool to characterize, discriminate, and better understand the formation mechanism of the different structures encountered during the formation of PECs.

## 1. Introduction

The complexation of oppositely charged polyelectrolytes (PEs) in aqueous solutions is a widespread associative process found in natural and man-made systems, which takes place through mainly cooperative electrostatic interactions. Complementary interactions like short-range hydrogen-bonding or long-range hydrophobic effect can also participate in the complexation process. Such PE complexes or PECs have endless applications in different technological fields ranging from water treatment [1], paper making [2], and food industry [3] to pharmaceuticals [4], cosmetics [5], tissue engineering [6], and biomedicine [7]. The formation of PECs involves two distinct steps: the generation of primary PECs by ion pairing, followed by their aggregation/reorganization into larger structures [8]. At stoichiometry, when the amount of anionic and cationic charges are equal, two very different situations can occur depending on the intensity of the interaction. When the charged chains are strongly interacting (binding constant k_b_ > 10^6^ M^−1^), the system undergoes a liquid–solid transition generating aggregates that eventually sediment with time as in the case of the PDADMAC/PSSNa system [9,10]. When the interaction is weaker, the system undergoes a liquid–liquid phase transition generating highly hydrated and dense coacervate droplets in equilibrium with the surrounding continuous phase depleted in macromolecules as in the PDADMAC/PANa system (k_b_ ~10^3^–10^4^ M^−1^) [11,12,13]. Off the stoichiometry, colloidal polyelectrolyte complexes are formed with a solvation, charge, hydrophobicity that vary with the molar charge ratio Z ([-]/[+]) [11,12,13].

The adsorption at the air/water interface of individual synthetic and natural polyelectrolytes, the main components of PECs, has been fairly well documented in the literature for several decades, particularly with the sodium poly(styrene sulfonate) (PSSNa) [14,15,16,17,18,19,20,21,22,23,24]. The adsorption of PEs differs in many respects from that of neutral polymers. Highly charged PEs are not hydrophobic enough to be surface active and therefore do not spontaneously adsorb at water–air interfaces at low concentrations [15,16,22]. In the presence of added electrolytes (salt) or at a sufficiently high concentration, they nevertheless adsorb with a much slower kinetics (a few hours) than in the case of neutral polymers, where equilibrium is reached in a few minutes. A well-known effect due to the presence of an electrostatic barrier is that the first adsorbing PE chains generate a negative adsorption potential that slows down the subsequent adsorption of additional charged chains. In this context, the addition of salt increases the ionic strength of the solution, screening the electrostatic interaction and resulting in a larger and faster decrease in surface tension. At a sufficiently high salt concentration, the surface layer of PEs approaches that of neutral polymers. 

If we add a second component of opposite charge in the solution, the surface activity properties of PEs can be completely different. A lot of work has been done in the last three decades to study the properties of PE/surfactant complexes (PESCs) at interfaces. Compared to neutral polymer/surfactant mixtures where only weak hydrophobic interactions can develop, PESCs have shown the coexistence of strong electrostatic and hydrophobic interactions [25,26,27]. This results in a complex but very rich adsorption pattern that has led to many important industrial applications ranging from enhanced oil recovery [28] and wastewater [29] to pharmaceuticals [30] and cosmetics [5]. Surface tension measurements of surfactants in the presence and absence of polymer were found to be particularly relevant to highlight the formation of specific interactions, if any, between the components [31].

Furthermore, if a PE replaces the surfactant, surprisingly few studies in the literature have focused on the interfacial properties of such PE complexes. Bago et al. recently highlighted the possibility of using PECs consisting of PDADMAC/PSSNa (solid-like colloidal PECs) and PDADMAC/PANa (near-neutral coacervate droplets) to stabilize oil/water emulsions while individual PEs are not emulsifiers because macroscopic phase separation occurs immediately after mixing [32,33]. The surface activity of PSSNa/PAH and PDADMAC/PSSNa/lipase PECs were also highlighted by Owiwe et al. and Generalova et al., respectively [20,34]. But in these very interesting works, it is difficult to know precisely which structures are responsible for these effects. Indeed, PECs can exist in different physical forms (soluble or insoluble colloidal complexes, coacervated droplets), which must have an impact on their surface activity properties, the objective of the study is precisely to establish this correlation. This will allow us to better use PECs to develop and modify the interfacial properties of various biphasic systems.

In this work, we systematically correlated the surface activity of PECs obtained at different molar charge ratios (Z) with the PEC structure determined by light scattering, zeta potential, and microscopy. Two model systems were examined, the weakly interacting PDADMAC/PANa system forming liquid-like PECs and the strongly interacting PDADMAC/PSSNa pair forming solid-like PECs (Figure 1). Our approach also showed that simple surface tension measurements at the air/water and water/water interface can be a very sensitive tool to characterize, discriminate, and better understand the formation mechanism of the different structures encountered during the formation of PECs.

## 2. Experimental Section

**Polyelectrolytes.** Poly (diallyldimethylammonium chloride) (PDADMAC, M_w_ ~ 100,000–200,000 g·mol^−1^) was purchased from Aldrich, Saint-Louis, MI, US, as 20 wt. % solution in water (Lot # 03530MS). Poly (acrylic acid sodium salt) (PANa, M_w_ ~ 2000 g·mol^−1^) and Poly (sodium 4- styrenesulfonate) (PSSNa, M_w_ ~ 70,000 g·mol^−1^) were purchased from Aldrich, Saint-Louis, MI, US, as powders (Lot # BCBF7673V) and (Lot # LR0001445584), respectively. Before use, the three polyelectrolytes were purified by extensive dialysis against pure water and then freeze-dried. Individual stock solutions were prepared at a concentration of 1 M (in repetitive units) in DI water by taking into account the water content as determined by thermogravimetric analysis (Q50, TA instrument). The pH of each solution was set to 10 with the help of 1 M sodium hydroxide. At such pH, the weak PANa polyanion was fully charged [35] as were the strong PDADMAC and PSSNa polyelectrolytes with a pH-independent charge.

Then, the stock solutions were diluted to the desired concentrations expressed in mM of repetitive units (0.62 mM, 6.2 mM, 18.6 mM, 62 mM, 434 mM, 930 mM) and the pH adjusted again to 10 (the volume of NaOH was taken into account for the final concentration). All solutions were filtered through 0.20 μm pore size cellulose acetate membranes (Millipore) before use.

**Preparation of polyelectrolyte complexes (PECs).** The PECs were formed at different molar charge ratios (Z) defined as $Z\;=\frac{\;\left[\mathrm{PANa}\right]\;}{\left[\mathrm{PDADMAC}\right]}$ or $Z\;=\frac{\;\left[\mathrm{PSSNa}\right]}{\;\left[\mathrm{PDADMAC}\right]}$ by considering the molar concentrations of repetitive units of respective PEs. PE solutions at 18.6 mM were used, so that the total PE concentration was also 18.6 mM whatever the Z value targeted. This relatively low polymer concentration was chosen to minimize aggregation phenomena. The PECs were prepared by injecting a certain volume of PDADMAC solution into a vial containing either a PANa or a PSSNa solution. The suspensions were thoroughly stirred for 2 min and rapidly used for DLS and surface tension measurements to avoid any significant sedimentation phenomena. The supernatants of PEC suspensions obtained at Z = 1 were also collected and analyzed after 2 weeks. 

**Dynamical light scattering (DLS) and ζ-potential.** Dynamic light scattering (DLS) experiments were performed by using an ALV laser goniometer, with a 22 mW linearly polarized laser (632.8 nm HeNe) and an ALV-5000/EPP multiple tau digital correlator with a 125 ns initial sampling time. The DLS autocorrelation functions were obtained at a scattering angle of 90° and a temperature of 20 °C. The intensity-weighted relaxation time (τ) distributions were extracted from the autocorrelation data using the multi-exponential CONTIN method. The apparent diffusion coefficients (D_app_) were obtained by considering D_app_ = (1/τq^2^) where τ is the mean relaxation time and q the scattering vector. The hydrodynamic radii (R_H_) were determined using the Stokes–Einstein relation, $R_{H}=\frac{k_{B}T}{6\pi \eta D}$ where k_B_ is the Boltzmann constant, T the temperature, η the viscosity of the solvent, and D the diffusion coefficient. ζ potential measurements were performed with a Zetasizer Nano-ZS (Malvern Instruments, Malvern, UK) at an angle of 173° using a 4.0 mW He-Ne laser operating at a wavelength of 632.8 nm. An automatic titrator MPT-2 (Malvern Instruments) was coupled to the Zetasizer allowing successive injections of one PE (titrant) into the second (titrated). The electrophoretic mobility of each solution was measured and converted to ζ potential (mV) by the Smoluchowski approximation model using the DTS program (Malvern).

**Small-angle X-ray scattering (SAXS)**. SAXS profiles were acquired on a XEUSS set-up (Xenocs, Grenoble, France) with a microfocus copper anode source, a scatterless collimation system, and a PILATUS 2-D detector, giving access to scattering wave vector q values from 0.009 to 0.5 Å^−1^. The PEC and individual PEs were put in thin glass capillaries. The resulting 2D images were found to be isotropic, and the data were azimuthally averaged to give the intensity scattering curve I(q), corrected for the experimental background, as a function of q = (4π/λ)sin θ, where λ = 0.154 nm is the wavelength of the Cu Kα radiation and θ is half the scattering angle. 

**Dynamic air–water surface tension measurements.** The dynamic surface tension of the PEC solutions was measured using a pendant-drop set-up (Tracker, from Teclis Scientific, Civrieux-d’Azergues, France). A drop of water of about 10 µL was generated with a syringe equipped with a Teflon needle. The variation of the surface tension at the solution/air interface was then monitored for more than 500 min at 22 °C with a drop volume kept constant during the whole measurement. The time-dependent interfacial tension was deduced from the axisymmetric drop shape by fitting it with the Laplace equation. The pendant drop method is well suited for dynamic surface tension monitoring because (i) it facilitates the creation of a pristine air/water interface, (ii) it uses small volumes of solution, and (iii) it avoids the problems of probe cleaning in force-based methods (Du Noüy ring or Wilhelmy plate) that can affect the final results [36].

**Water–water interfacial tension measurements.** The dispersion of PECs obtained from 18.6 mM PE solutions at a charge ratio Z = 1 was placed in a 2 mm-thick quartz cell with parallel walls. A complete macroscopic liquid–liquid phase separation (coacervation) occurred within 2 days, giving rise to a net interface between the polymer-rich (bottom) and polymer-dilute (supernatant) phase. It is the capillary length scale defined as:(1)$$l_{c}=\sqrt{\frac{\gamma_{cc}}{\Delta \rho \times g\;}}$$
that sets this profile with Δρ the mass density difference between the two phases, g the gravitational acceleration and $\gamma_{cc}$ the (very low) interfacial tension between the dense complex coacervate (CC) and the macromolecule-depleted phase. $\gamma_{cc}$ was then computed by measuring Δρ and $l_{c}$. Δρ was measured using a Density Meter (DMA 4100M, Anton Paar, Graz, Austria) with a resolution of 0.0001 g·m^−3^. $l_{c}$ was obtained by analyzing the static interfacial profile near the vertical wall of the cell with a shape given by the generalized Young–Laplace equation connecting capillary pressure with curvature and surface tension for the case of a flat wall as [37,38]:(2)$$\frac{x}{l_{c}}=\lceil arccosh\left(\frac{2l_{c}}{y}\right)-arccosh\left(\frac{2l_{c}}{h}\right)+{\left(4-\frac{h^{2}}{l_{c}^{2}}\right)}^{1/2}-{\left(4-\frac{y^{2}}{l_{c}^{2}}\right)}^{1/2}\rceil$$
where x is the distance to the vertical wall, y the height of the profile above the flat level far away from the wall, and h is the meniscus contact height at x = 0. The (X,Y) profile was observed and recorded using an optical microscope (X5 objective) rotated by 90° equipped with a digital camera, computed using the ImageJ software and finally fitted to Equation (2). The left and right profiles on both sides of the cell were measured and averaged. 

## 3. Results and Discussion 

### 3.1. Surface Tension of Individual Polyelectrolyte Solutions

We first investigated the interfacial behaviour of the three individual polyelectrolyte solutions considered in this work prior to study the behaviour of PECs. Figure 2a shows the dynamic surface tension of PANa solutions at different concentrations. At low PANa concentrations (c ≤ 6.2 mM), almost no adsorption occurs due to the hydrophilic and charged nature of the polymer chains, as reported for various polyelectrolytes [14,15,16,17,18,19,20,21,22,23,24]. For higher PANa concentrations (c > 6.2 mM), the surface tension significantly decreased due to the concomitant increase of the ionic strength in the medium when the PE concentration increased. Indeed, the concentration of free sodium counterions scales with the PE concentration as [Na^+^]_free_ ~ C_PANa_ × *f_eff_*, where *f_eff_* is the effective charge density of the polyelectrolyte (*f_eff_* ~ 0.35 for PANa) in relation with the Manning–Oosawa condensation [39,40]. Therefore, by increasing the PANa concentration, the electrostatic repulsions between the carboxylate units become increasingly screened, thus favoring the polymer adsorption at the air-water interface. In this concentration regime, it can be observed that adsorption mostly occurs at short times (< 1 h), which is related to the diffusion of polymer chains, while at longer times, adsorption is much less important due to the formation of an electrostatic barrier generated by the previously adsorbed PANa chains at the air–water interface [16]. This is a general trend as shown by the adsorption behaviour of the two other PEs investigated (Figure 2b and Appendix A) and well in line with the literature [14,15,16,17,18,19,20,21,22,23]. It emphasizes that the affinity of PEs with the surface is mainly determined by the screening conditions rather than their chemical composition. Besides, it is important to note that the dynamic surface tension of the three PEs did not reach equilibrium values even after 500 min, as reported elsewhere for other PEs [16]. The small but still persistent adsorption at long times could be ascribed to the formation of metastable microdomains or aggregates at the interface, similar to those typically found in PE solutions [16]. We can then estimate the molar surface excess from the concentration dependence using the equilibrium Gibbs equation [15]: (3)$$\Gamma_{PE}=-\frac{1}{RT\left(1+Nf_{eff}\right)}\frac{d\gamma }{d\left(\ln c_{PE}\right)}$$
where *f_eff_*, *c_PE_*, and *N* are the effective charge density, concentration, and polymerization degree of the PE, respectively. The approximate surface excess was found to be 2.6 × 10^−8^ mol/m^2^ (1.2 mg/m^2^), 6.2 × 10^−7^ mol/m^2^ (1.25 mg/m^2^) and 2.2 × 10^−8^ mol/m^2^ (1.5 mg/m^2^) for PDADMAC, PANa, and PSSNa, respectively, a result suggesting that smaller chains of PANa 2 k can accommodate more easily than larger ones at the air/water interface. 

### 3.2. Dynamic Light Scattering Analysis and Zeta Potential of PECs

The PEC dispersions generated at different molar charge ratios (Z=[-]/[+]) from PE solutions at 18.6 mM were at first characterized by dynamic light scattering (DLS). Figure 3 shows the relaxation time spectra of the two systems obtained from correlation functions by applying an inverse Laplace transform. For individual PDADMAC, PSSNa, and PANa solutions, a fast mode was observed originating from the coupling of the PE motion with the dynamics of small and much faster counterions [41]. For the weakly complexing PDADMAC/PANa system, due to the small size of PANa chains, so-called soluble complexes were formed at low Z [11,12,42]. These structures were made of single PDADMAC chains complexed with few PANa ones. They carry a net positive charge as shown by zeta potential measurements (Figure 4) that is high enough to prevent self-association phenomena, as shown by the value of the hydrodynamic radius (R_H_) around 4–5 nm, well in line with the coil size of a PDADMAC chain in the presence of salt. It should be noted that soluble complexes represent a very specific kind of complex structure that can only be obtained under strict conditions of high chain length asymmetry between the two PEs [12,42]. At higher Z (~0.6), where most charges of PDADMAC are neutralized through ion pairing with oppositely charged PANa chains, the relaxation times abruptly increased above 1 ms corresponding to R_H_ ~ 180 nm. In that case, PDADMAC chains are no longer hydrophilic and can self-assemble into small particles through hydrophobic interactions. Particles are stabilized by the excess of PDADMAC at Z < 1 and by the excess of PANa at Z > 1. In the strongly complexing PDADMAC/PSSNa system, colloidal complexes were generated from low Z values due to a much strongly interacting system with respect to the larger molar mass and higher hydrophobicity of PSSNa. 

Furthermore, at charge stoichiometry (Z = 1), a fundamental structural difference appears between the two systems. As expected, the highly complexing PDADMAC/PSSNa system undergoes a liquid–solid phase transition leading to solid aggregates while the weakly complexing PDADMAC/PANa system undergoes a liquid-liquid transition leading to coacervate droplets; a feature clearly visible in the optical microscopy images in Figure 3, which also generates a maximum turbidity at Z = 1 just after mixing. It is also interesting to note that the suspension of PSSNa-based complexes off stoichiometry is more turbid compared to PANa-based complexes. It shows that larger and probably denser particles of complexes were obtained with PSSNa as polyanion even though this was not readily detectable on the DLS size distributions due to the size dispersity of the structures and the presence of slow modes [43].

The variation of the zeta potential (ζ) shows a typical charge inversion from positive ζ values when the polycation is in excess (Z < 1) to negative ones when the polyanion is in excess (Z > 1) (Figure 4) in line with the recent work of Bago et al. [32]. What is more interesting is the pattern of the variation and the absolute values of ζ. The charge inversion is sharper in the case of complexes containing PSSNa. Overall, this shows that complexes made of PSSNa have a strict charge separation between the core and shell, whereas for PANa-based complexes, the charges are more evenly distributed throughout the complex structure. This supports a complexation mechanism where the hydrophobicity of PSSNa favors the formation of dense and solid-like complexes with less possibility for structural rearrangements. A feature that often leads to charge neutralization (ζ~0) for values of Z slightly off the charge stoichiometry. On the contrary, complexes made from PANa are softer, better hydrated, and more prone to structural changes, thus favoring the transition from soluble complexes at Z < 0.6 to dispersed complexes at 0.6 < Z < 1 and coacervate droplets at Z close to 1. In this case, the chains can easily rearrange to maximize ion pairing given rise to neutralization at Z~1. In a study on the influence of the hydrophobicity of PEs in PEC formation, Mende et al. concluded that PEs containing hydrophobic styrenic units favor the formation of compact structures compared to less hydrophobic PEs that instead lead to swollen particles [44]. Higher hydrophobicity leads also in general to lower colloidal stability. Finally, as expected for such a system, no significant variation of the pH was observed during the complexation, which excludes all possibilities of acid–base reactions (data not shown).

### 3.3. Surface Tension of PEC Dispersions

DLS analyses showed the presence of various colloidal structures in the PEC dispersions prepared from PE solutions at 18.6 mM. There was no large difference in size between the complexes obtained from PDADMAC/PANa and PDADMAC/PSSNa at various Z ratios even though their physical nature differs, liquid-like for the former and solid-like for the latter. In the following, we will show that these PECs have a specific interfacial signature, even if individual PEs barely adsorbed to the air/water interface at a concentration of 18.6 mM (Figure 2). 

Figure 5 shows the time-dependent adsorption at the water/air interface of the different PECs generated as a function of the molar charge ratio (Z) at pH 10. For the two complex systems studied, the dynamic surface tension (γ) profiles are very different from those obtained with PEs alone (Figure 2). First, the γ values obtained with PECs are much lower than those found with PEs alone at similar concentration (the overall polymer concentration in PEC suspension was 18.6 mM for all Z values) (Figure 5a,b). This highlights a more efficient charge screening of PDADMAC by complexation with a polyanion rather than by an increase of the ionic strength. Figure 5c shows that γ values obtained after 500 min decreased as the system gets closer to stoichiometry (Z = 1) for both PDADMAC/PANa and PDADMAC/PSSNa complexes whenever the complexation takes place in presence of an excess of PDADMAC (Z < 1) or an excess of polyanion (Z > 1). This agrees well with an increased level of charge neutralization of PEs forming increasingly hydrophobic surface-active species. The fact that the variation of γ is symmetrical around Z = 1 also emphasizes that the interfacial activity of both systems depends greatly on the level of complexation and little on the positive or negative nature of the charges in excess at PEC surface. The behavior at stoichiometry (Z = 1) was particularly informative of the physical nature of the complexes. For PDADMAC/PANa at Z = 1 where the system undergoes a liquid–liquid phase transition forming coacervate microdroplets, the surface tension of the PEC suspension at Z=1 just after complexation is similar to that measured in the supernatant after sedimentation and coalescence of the droplets (Figure 5a, symbol 1-sup). This shows that the surface-active species must be similar in both phases. Therefore, it rules out the possibility for large coacervate droplets to participate in the adsorption process. In fact, the adsorbing species are likely free macroions or small complex particles in equilibrium with the coacervate droplets, in agreement, respectively, with the segregation mechanism proposed by Veis and the intercomplex disproportionation described by Shklovskii et al. [45,46]. The two scenarios were reviewed by Kizilay et al. [47]. The fact that the surface tension drops quite rapidly with time at Z = 1 in the PEC suspension and supernatant (Figure 5a) is also in line with the diffusion and adsorption at the liquid–air interface of small complexes or free macroions. For Z values different from 1, the adsorption kinetics of complexes is slower because of their larger size and higher charge density (Figure 4 and Figure 5a). 

The interfacial behavior of PDADMAC/PSSNa was clearly different. First, the adsorption of complexes at the interface was much slower than for PDADMAC/PANa, especially for charge ratios far from stoichiometry (Z = 0.1, Z = 0.2 and Z = 10), as judged from the variations of γ (Figure 5b). The surface tension values after 500 min were also higher for PSSNa-based complexes (Figure 5c). These trends reflect the behaviour of rather large, dense, and highly charged particles (Figure 4) with less affinity with the water-air interface. In agreement with previous studies [44], it confirms the solid-like nature of PSSNa-based complexes in contrast to PANa-based complexes which are softer, more hydrated and thus with interfacial properties closer to those of polymer chains. It has been recently shown that solid particles have in general low interfacial affinity while particles decorated with polymer chains can adsorb at the water–oil interface depending on the hydrophobicity of the polymer. For example, latex particles decorated with polyamine chains can adsorb at the interface at pH 10 because amine groups are neutralized under such conditions [48]. PDADMAC/PSSNa complexes have such a core-shell structure with a dense and solid hydrophobic core resulting from the segregation of complexed segments and a stabilizing shell of excess PE. Then, the affinity of the shell with the water–air interface must depend greatly on the degree of complexation. 

For Z close to 1 (Z = 0.6, Z = 1.7), a significant fraction of complex particles sedimented due to their large size and poor colloidal stability (Figure 3b). Under these conditions, the few remaining particles capable of adsorbing at the interface were probably small and poorly charged, which would then explain the rapid and relatively large decrease in surface tension (Figure 5b). For Z = 1, a fraction of the neutral complexes adsorbed rapidly and massively at the interface as seen by the variation of γ. Even though the equilibrium value of γ was not reached after 500 min, one can figure out the final state of the system. Indeed, the γ value in the supernatant after complete equilibration of complexes (2 weeks) was close to that of water (72 mN/m) (Figure 5b), which means that PSSNa-based complexes have fully sedimented. It also implies that complexes were not in an equilibrium state in contrast to complexes obtained from PANa where free macroions or soluble complexes remain in solution. This is in agreement with a complete liquid-solid phase separation for the PDADMAC/PSSNa system. 

We have shown so far that the measurement of the surface tension can be a very useful tool to better understand the formation mechanisms of PECs and to detect in a very sensitive way the presence of a very small amount of aggregates and other precursors in the supernatant phase. In Figure 5c, the value of Z = 1 was obtained by simple manual mixing of two equimolar solutions of PEs of opposite charge. Is this method accurate enough to determine the exact charge stoichiometry? To answer this question, we slightly varied the stoichiometry of the mixture around Z = 1 and measured the surface tension of the supernatant to verify that it was indeed minimum. 

Several insights can be drawn from data plotted in Figure 6. First, there is a value of Z equal to 1 that maximizes the surface tension of both supernatants and decreases symmetrically on both sides. This value can be considered as the experimental “true” charge stoichiometry. It is remarkable that the surface tension varies so strongly over such a small range of Z, thus emphasizing that this method is very efficient to determine precisely the stoichiometry in PECs systems. In particular, the method is more discriminating than turbidity measurements, which would yield optically transparent supernatants around Z = 1. It is also remarkable that the complexation stoichiometry was 1 despite the structural differences of the PEs. In fact, the PEs adopt an open extended conformation in salt-free solution, which favors the juxtaposition of long portions of oppositely charged PEs. Then, according to Michaels et al., the high local concentration of microions and the slow diffusion of the released microions from the sites of reaction allow sufficient charge screening so that rearrangements of rotational conformations can take place and favor the complete ionic pairing between PEs [49].

Furthermore, the surface tension measured at Z = 1 for the aggregating PDADMAC/PSSNa system was similar to that of pure water, suggesting that all PE chains were indeed engaged in the formation of dense and solid aggregates during the liquid-solid phase transition, with the release of all counterions into the supernatant. This hypothesis was verified by measuring the conductivity of the supernatant at Z = 1 and of NaCl solutions prepared at various concentrations, as proposed a long time ago by Michaels et al. on a similar PEC system [49]. The conductivity in the supernatant corresponded to a NaCl concentration of 8.75 ± 0.51 mM, which is in good agreement with the complete release of counterions from PSS(Na+) and PDADMAC(Cl-) solutions prepared at 18.6 mM and mixed in equal volume, the theoretical NaCl concentration being then 9.3 mM. 

For the PDADMAC/PANa system, the surface tension of the supernatant (at Z = 1.0) clearly indicates that the supernatant contains surface-active species in equilibrium with the coacervate droplets, as discussed previously. In order to know more precisely the structure of these species, the supernatant was analyzed by SAXS (Figure 7). From the SAXS point of view, it appears that the supernatant contains little or no objects in solution while the interfacial tension measurements indicate the presence of surface-active species. It cannot be the PEs alone, as it would require at least a concentration of 100 mM (see Figure 2) to lower the surface tension to 64 mN/m, which would then be visible in SAXS. We then prepared PDADMAC/PANa PECs out of stoichiometry at a low concentration of 1 mM where neither SAXS nor DLS gave a consistent signal. Dynamic surface tension measurements performed on these diluted solutions (Appendix A) showed a slight decrease in surface tension for the three Z studied. These data suggest then that the supernatant probably contains very small numbers of PECs in equilibrium with the coacervate phase, an appealing hypothesis that needs to be confirmed by a dedicated and comprehensive study on the subject. 

### 3.4. Interfacial Tension between the Dilute and Concentrated Coacervate Phase 

At this point, all PEC dispersions were analyzed by surface tension measurements with the exception of the liquid–liquid interface formed during the complex coacervation of the PDADMAC/PANa system at Z = 1. The interfacial tension $\gamma_{cc}$ between the dilute and the concentrated phases coexisting in the complex coacervate can be 10,000 times lower than that of pure water, on the order of 1–100 µN/m, ref [50,51,52,53,54] and thus quite difficult to measure directly after complexation. Within 14 days (or with the help of centrifugation) however, a complete macroscopic liquid–liquid phase separation is usually occurring giving rise to a net interface between the dense and light phase. In this particular geometry, the interfacial tension $\gamma_{cc}$ can be accurately measured if the capillary length *l_c_* as well as the density difference Δ*ρ =*
*ρ*_2_
*−*
*ρ*_1_ between the two phases is known (see material and method) [37]. *l_c_* was obtained here by fitting the static interfacial profile to the generalized Young–Laplace equation near a flat wall, as shown in Figure 8. *γ_cc_* was found to be 311 µN/m in agreement with values found in the recent literature on water/water interfacial tension in both associative complex coacervates [50,51,52,53,54] and segregative aqueous two phase systems (ATPS) (dextran/PEG [55], dextran/gelatin [56]). A very small amount of energy is then sufficient to create an additional surface between the two coexisting PDADMAC/PANa phases; no ion pair breaking is then necessary but only their redistribution in the case of a neutral complex [57]. This low energy barrier will facilitate the transfer of actives into the denser dispersed phase. This key property is at the origin of the celebrated encapsulation/sequestration properties of these water-in-water emulsions [58]. It is therefore very important for all applications to be able to stabilize these interfaces in order to avoid any short-term macroscopic phase separation [59]. 

## 4. Conclusions

In this work, we highlighted that simple surface tension measurements at the air/water and water/water interface can be a very sensitive and discriminating tool to study the structure and formation mechanisms of polyelectrolyte complexes. This approach allows to finely characterize the different structures formed during the “electrostatic” complexation of polyelectrolytes of opposite charge as a function of the molar charge ratio Z = [-]/[+] and the interaction strength. For the strongly and weakly interacting PE systems studied in this work, the surface tension of the complexes decreases as the system approaches charge stoichiometry (Z = 1) whenever the complexation occurs in the presence of excess PDADMAC (Z < 1) or excess polyanion (Z > 1). A scenario consistent with an increased level of charge neutralization of PEs forming increasingly hydrophobic and neutral surface-active species, in agreement as well with zeta potential data. 

In addition, the behavior at stoichiometry (Z = 1) was particularly informative about the physical nature of the complexes. In the PDADMAC/PANa system that undergoes a liquid–liquid phase transition at Z = 1 (formation of coacervate microdroplets), the surface tension of the suspension just after complexation is equal to that measured in the supernatant after macroscopic phase separation, suggesting that the surfaces-active species must be similar in both phases; a feature consistent with a system at equilibrium. In the PDADMAC/PSSNa system at Z = 1, the surface tension of the supernatant after two weeks was close to that of water, suggesting that the PSSNa-based complexes have completely sedimented, in contrast to the PANa-based ones where soluble complexes remain in solution; a feature consistent with the complete liquid–solid phase separation of a non-equilibrium system.

In addition, the high sensitivity of surface tension measurements, which can detect the presence of trace amounts of aggregates and other precursors in the supernatant while turbidity measurements yield optically transparent supernatants, allows for very accurate determination of the exact or “true” charge stoichiometry of the complexes. 

To finalize the characterization of the weakly interacting PDADAMAC/PANa system, the water/water interfacial tension between the dilute phase (supernatant) and the concentrated phase of the coacervate was measured using the generalized Young–Laplace equation. A value of ~311 µN/m was found in agreement with other macromolecular systems developing an ultra-low water/water interfacial tension (complex coacervates, biocondensates, and ATPS). This property allows these water-in-water (W/W) emulsions to efficiently encapsulate actives of interest in the dispersed phase but also makes them very difficult to stabilize [59,60]. 

## Figures and Tables

**Figure 1 polymers-13-03848-f001:**
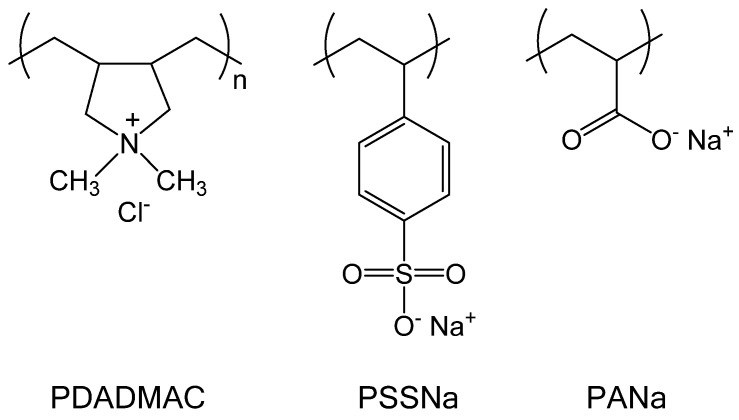
Chemical structure of PDADMAC, PSSNa, and PANa.

**Figure 2 polymers-13-03848-f002:**
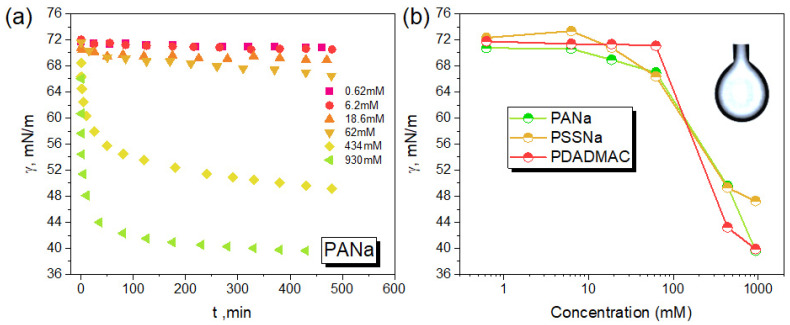
Dynamic surface tension (γ) of PANa solutions at different concentrations (in repetitive units) determined by pendant-drop measurements (**a**). Variation of the surface tension as a function of the concentration of the polyelectrolyte solution (PANa, PDADMAC, PSSNa) (**b**). Data were collected during 500 min.

**Figure 3 polymers-13-03848-f003:**
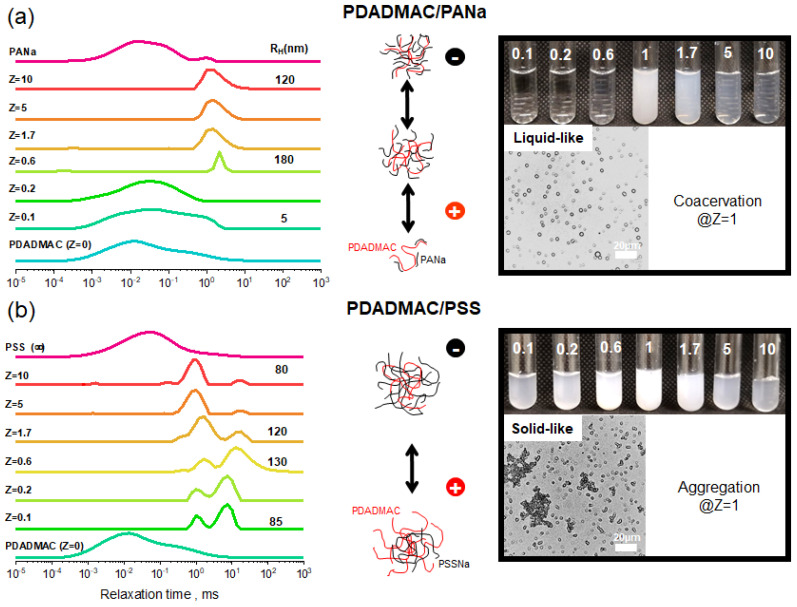
Spectra of the relaxation times (and hydrodynamic radii) and photographs of the PEC dispersions taken at different Z values together with microscopic images taken at charge stoichiometry (Z = 1) of PDADMAC/PANa (**a**) and PDADMAC/PSSNa. (**b**) PECs were generated at different molar charge ratios (Z) from PE solutions prepared at 18.6 mM, pH 10 without added salt.

**Figure 4 polymers-13-03848-f004:**
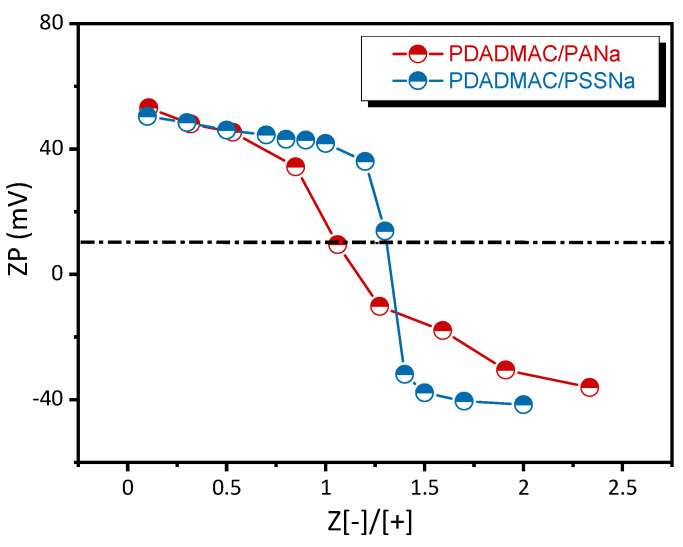
Zeta potential (ζ) of PECs prepared at various charge ratios (Z): PDADMAC/PANa and PDADMAC/PSSNa. PECs were prepared from PE solutions at 18.6 mM, pH 10 without added salt.

**Figure 5 polymers-13-03848-f005:**
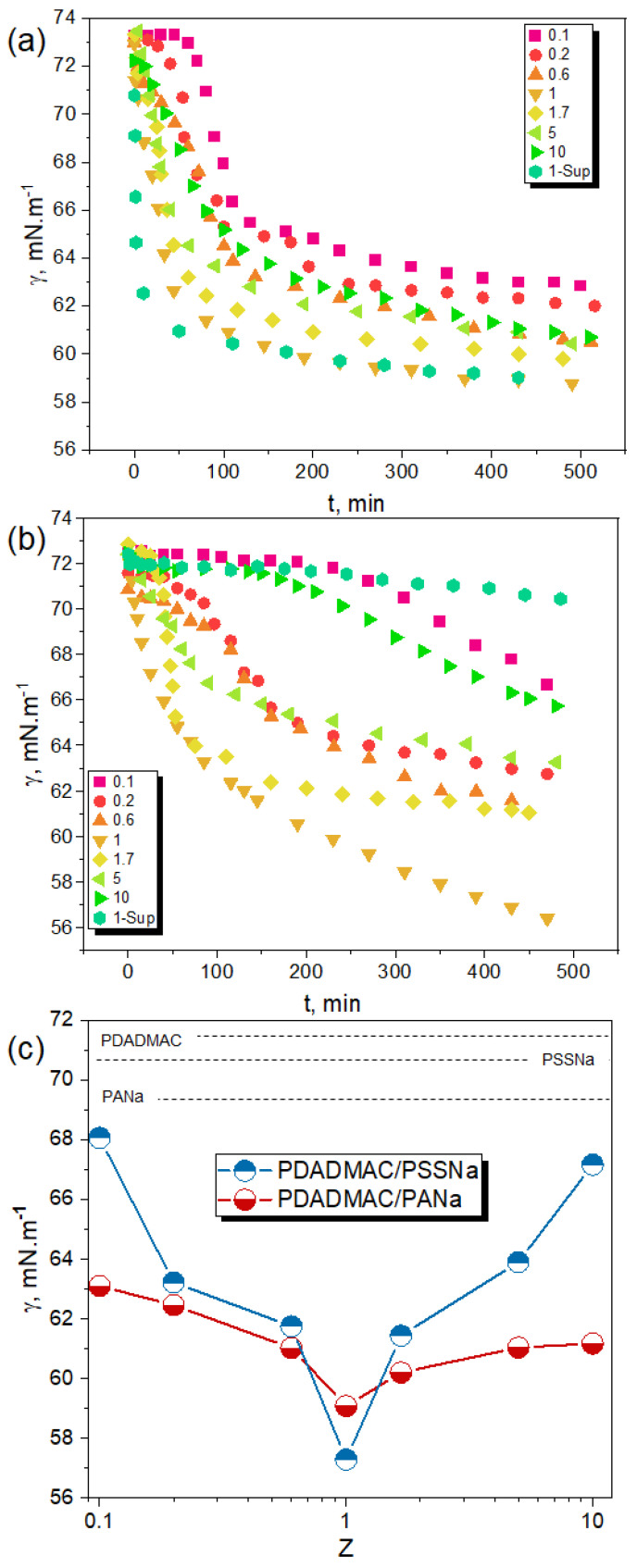
Dynamic surface tension (γ) of PDADMAC/PANa (**a**) and PDADMAC/PSSNa (**b**) PEC suspensions at different molar charge ratios (Z). Variation of γ obtained after 500 min as function of Z for the two PEC systems (**c**). PECs were prepared from PE solutions at 18.6 mM and pH 10 and rapidly analyzed after preparation. 1-sup in the (**a**,**b**) legend refer to the surface tension (γ) of the supernatant after the macroscopic phase separation occurring at Z = 1 for both systems. Dashed lines correspond to the surface tension of individual PE.

**Figure 6 polymers-13-03848-f006:**
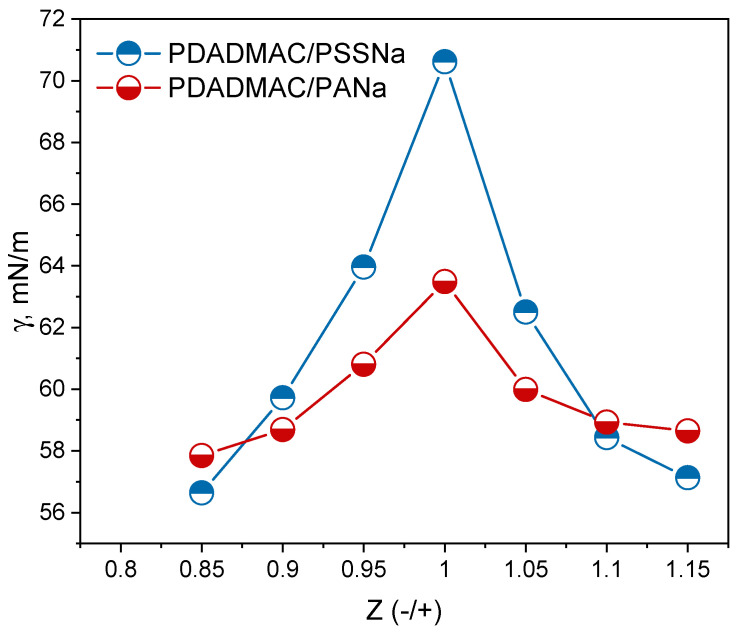
Surface tension (γ) measurements of the supernatant of PEC dispersions at different charge ratios (Z) around the charge stoichiometry (Z = 1) for PDADAMAC/PANa and PDADMAC/PSSNa. PECs were prepared from 18.6 mM PE solutions, the supernatants were collected after a complete macroscopic phase separation (2 weeks), and the surface tension was measured after 500 min when the signal became constant.

**Figure 7 polymers-13-03848-f007:**
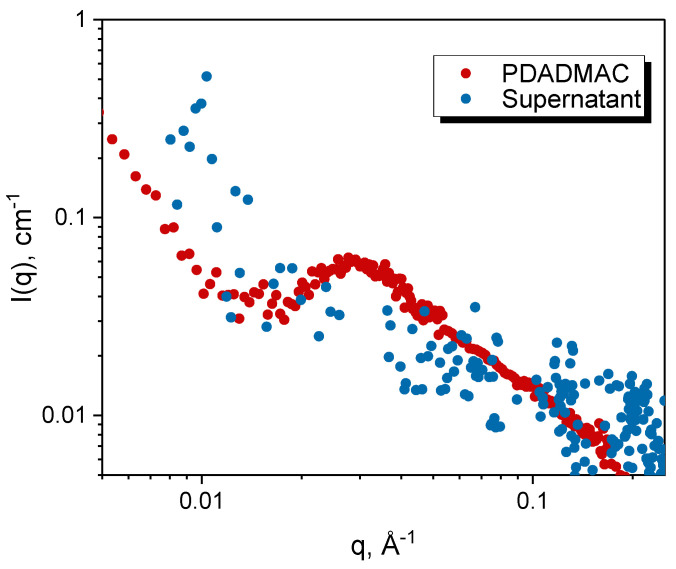
SAXS analysis of the supernatant phase of the PDADMAC/PANa system prepared at Z = 1 from 18.6 mM PE solutions. The signature of a 18.6 mM PDADMAC solution is also shown for comparison. We also note the presence of a polyelectrolyte peak (q~0.28 Å^−1^) expected for this type of charged macromolecule.

**Figure 8 polymers-13-03848-f008:**
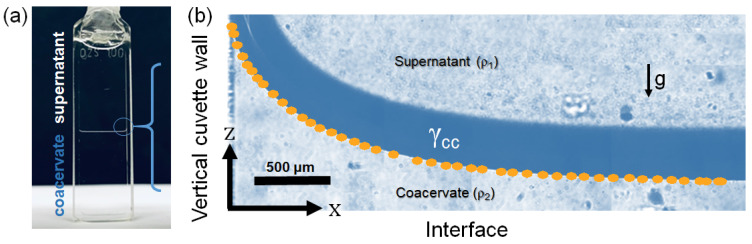
(**a**) Macroscopic phase separation obtained after 14 days after complexation of the PDADMAC/PANa system prepared from 18.6 mM PE solutions at charge stoichiometry (Z = 1). A clear separation between the dense coacervate phase and the macromolecule-depleted supernatant is visible. (**b**) Optical microscope image of the liquid–liquid interface near to the cell wall. The static interfacial profile was fitted (orange dots) to the generalize Laplace–Young equation (see Equation (2)) to extract the capillary length scale *l_c_* to compute the interfacial tension *γ_cc_*.

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
