# Peer review of "Interfacial Behavior of Solid- and Liquid-like Polyelectrolyte Complexes as a Function of Charge Stoichiometry"

_polymers, 2021, doi:10.3390/polym13213848_

Round 1

Reviewer 1 Report

I recommend this manuscript for publication after minor revision, which is desired at following points:

The PANa and PSSNa used in the study have Mw of 2000 g/mol and 70000 g/mol, respectively.

Why didn’t you use polymers with similar molecular weight? Can they rule out that the differences in behavior are due to the large differences in molecular weight?

Page 7, line 261:

“Finally, no significant variation of the pH was observed during the complexation which excludes all possibilities of acid-base reactions (data not shown).”

Can you explain in more detail what acid-base reactions might be expected? The polyacrylic acid is fully dissociated at pH 10. What can happen at pH 10?

Typo

Page 2, line 93 and 94:

Shortcut for poly(acrylic acid sodium salt): PAANa or PANa?

Shortcut for poly (sodium 4-styrenesulfonate): PSS or PSSNa?

Page 5, line 183:

“Therefore, by increasing the PANa concentration, the electrostatic repulsions between the sulfonate units are screened …”

Replace sulfonate by carboxylate.

Reviewer 2 Report

The presented paper emphasizes the use of simple surface tension measurements at the air/water and water/water interface as a very sensitive and discriminating tool to study the structure and formation mechanisms of polyelectrolyte complexes. The performed studies and measurements are appropriate and well described. Moreover, the discussion is overall well conducted. I believe that after the suggested changes, this work should be published in the Polymers journal.

Comments:

  1. Introduction:
  • Please improve the quality of the citations in the manuscript: add space before making the citation (e.g. line 37). Moreover the full stop should be after the citation, not before (e.g. line 40).
  • Please provide full chemical name of the used polyelectrolytes before the first abbreviation.
  • Lines 33-35: The complexation of oppositely charged polyelectrolytes (PEs) in aqueous solutions is a widespread associative process found in natural and man-made systems which takes place through mainly cooperative electrostatic interactions. As the Authors discuss other interactions present in the studied systems (line 225) please expand this part of the manuscript.
  • Lines 52-53: We can obtain an overview of PE adsorption from one of 52 the most studied systems to date: the sodium poly(styrene sulfonate) (PSSNa). Please either elaborate this though or delete.
  • Lines 65-72: I believe that this paragraph could be further expanded. Interesting works describing the properties and characterization, as well as the adsorption characteristics of PESCs should be added [[1]-[2][3]].
  • Lines 73-92: This paragraph is important from the point of view of this paper, but I do not think that it is sufficiently described. Please provide more thorough literature review regarding the subject of PECs.
  • Line 82: If the question is risen here, shouldn’t the Authors at least try to take up the subject?
  • Lines 83-89: What about the other methods that were used in the following study, e.g. SAXS and DLS?
  • In my opinion an additional paragraph describing the influence of environmental factors such as pH and the presence of salt on the properties of PECs should be added.
  1. Experimental
  • Line 93: Either Poly(acrylic acid sodium salt) should be abbreviated here as PANa, or the abbreviations in the whole discussion should be changed to PAANa. Please check and correct.
  • Line 102: What was the reason behind choosing such concentration? Please explain briefly.
  • Lines 113-114: How does the sedimentation affect the measurements? During the DLS and ST measurements that takes quite some time, the sedimentation can cause the change in the measured parameters.
  • The information about pH of the studied systems should be added to the experimental section.
  1. Results and discussion
  • Lines 178-180: For higher PANa concentrations (c > 6.2 mM), the surface tension significantly decreased due to the increase of the ionic strength in the medium. I feel like this is too much of a simplification. The decrease of the ST is, as described later, the result of the increased adsorption of the PE at the air/water interface. Please rewrite this sentence.
  • Figure 2: Please include the a) and b) marks on the figure, so it is easier to differentiate between them when reading the discussion part.
  • Lines 217-221: These structures were made of single PDADMAC chains complexed with few PANa ones. They carry a net positive charge that is high enough to prevent self-association phenomena, as shown by the value of the hydrodynamic radius (RH) around 4-5 nm, well in line with the coil size of a PDADMAC chain in presence of salt (see Figure 7). Adding the reference to the zeta potential (ZP) data would be helpful for readers out of the subject because they may not know how the Authors obtain the information about the net charge.
  • Lines 246-248: If I understand the Authors correctly, the ZP in the stoichiometry (Z=1) should be close to 0. However, this is possible only if the total polymer chain charge of the PE is similar. As we all well known, it is not common. The Authors should also explain why the ZP is higher in the PDADMAC/PSSNa system when molar charge ratio Z=1. There are several reasons behind this phenomenon that should be at least mentioned. For example, including the information about the net charge (ZP) of the used PEs in pH=10 would allow better understanding of the situation. Please extend the subject so the readers out of the scope can understand the problem.
  • Line 263: If it is possible, I think that the data should be included in the supplementary material section.
  • Lines 278-281: First, the γ values obtained with PECs are much lower than those found with PEs alone at similar concentration (the overall polymer concentration in PEC suspension was 18.6 mM for all Z values) (Figure5a and 5b). Perhaps the synergistic effect in decreasing the surface tension should be discussed here?
  • Figure 5: Please include the a), b) and c) marks on the figure, so it is easier to differentiate between them when reading the discussion part. Moreover, why is the γ of PANa and PSSNa shown at Z=10, whereas for PDADMAC it is at Z=0,1. I believe that this should be shown at Z=1?
  • Lines 320-322: It has been recently shown that solid particles have in general low interfacial affinity while particles decorated with polymer chains can adsorb at the water-oil interface depending on the hydrophobicity of the polymer. I doubt that PECs that are considered as soft matter materials should be compared to “solid” particles. If the Authors believe that PECs could be used in the area of the pickering emulsions please provide more detailed review. Otherwise please rephrase this part.
  1. Conclusions
  • Lines 458-461: A very low value of ~5.2 μN/m in agreement with the very low water/water interfacial tension that develops at the water/water interfaces in complex coacervates, bio condensates and ATPS; a key property to efficiently encapsulate/sequester actives in the denser dispersed phase; but also challenging to stabilize. This part is unclear, please rephrase.

[1] La Mesa C. (2005) Journal of Colloid and Interface Science 286, 148-157; https://doi.org/10.1016/j.jcis.2004.12.038

[2] Grządka E., Matusiak J., Stankevič M. (2019) Journal of Molecular Liquids 283, 81-90; https://doi.org/10.1016/j.molliq.2019.03.059

[3] Grządka E., Matusiak J., Godek E., Maciołek U. (2021) Journal of Molecular Liquids 343, 117677; https://doi.org/10.1016/j.molliq.2021.117677

Round 2

Reviewer 2 Report

The Authors carefully checked and corrected the original manuscript following my concerns and questions. Therefore, I am satisfied with the changes that were made. I recommend to publish the paper titled “Interfacial behavior of solid- and liquid-like polyelectrolyte complexes as a function of the charge stoichiometry”’ in the Polymer journal.